# An Ecological Dynamics Approach to Understanding Human-Environment Interactions in the Adventure Sport Context—Implications for Research and Practice

**DOI:** 10.3390/ijerph19063691

**Published:** 2022-03-20

**Authors:** Tuomas Immonen, Eric Brymer, Keith Davids, Timo Jaakkola

**Affiliations:** 1Faculty of Sport and Health Sciences; University of Jyväskylä, 40014 Jyväskylä, Finland; timo.jaakkola@jyu.fi; 2Faculty of Health, Southern Cross University, Gold Coast Campus, Lismore, NSW 2480, Australia; eric.brymer@scu.edu.au; 3Sport & Human Performance Research Group, Sheffield Hallam University, Sheffield S1 1WB, UK; k.davids@shu.ac.uk

**Keywords:** adventure sport, extreme sport, ecological dynamics, transdisciplinary, form of life, skill, skill development, decision-making, freeriding, avalanche education

## Abstract

The last few decades have witnessed a surge of interest in adventure sports, and has led to an emerging research focus on these activities. However, recent conceptual analyses and scientific reviews have highlighted a major, fundamental question that remains unresolved: *what* constitutes an adventure sport. Despite several proposals for definitions, the field still seems to lack a shared conceptualization. This deficit may be a serious limitation for research and practice, restricting the development of a more nuanced theoretical explanation of participation and practical implications within and across adventure sports. In this article, we address another crucial question, *how* can adventure sports be better understood for research and practice? We briefly summarize previous definitions to address evident confusion and a lack of conceptual clarity in the discourse. Alternatively, we propose how an ecological perspective of human behaviors, such as interactions with the environment, may provide an appropriate conceptualization to guide and enhance future research and practice, using examples from activities such as freeride skiing/snowboarding, white-water kayaking, climbing, mountaineering and the fields of sport science, psychology and avalanche research and education. We draw on ecological dynamics as a transdisciplinary approach to discuss how this holistic framework presents a more detailed, nuanced, and precise understanding of adventure sports.

## 1. Introduction

The last few decades have witnessed a surge of interest in adventure sports, and an emerging research focus, especially on the psychological and emotional aspects of these activities for health and wellbeing. However, recent conceptual analyses and scientific reviews [1,2,3,4], have highlighted a major, fundamental question that remains unresolved: *what* constitutes an adventure sport? Despite several proposals for definitions, the field still seems to lack a shared conceptualization. This deficit may be a serious limitation for research and practice, restricting the development of a more nuanced theoretical explanation of participation and practical implications within and across adventure sports. To help us resolve this issue, here, we propose how adventure sports can be conceptualized using the *ecological dynamics* framework [5].

Traditional theoretical approaches have focused on participation in adventure sports which has been described as *dangerous*, *reckless*, *unhealthy*, or *harmful* to participants [3,4,6]. Research in the psychology of adventure sports has revealed an ‘organismic asymmetry’ or bias towards seeking internalized explanations for behavior within an individual (for examples in sport science and psychology, see [7,8]). An organismic asymmetry may emphasize the role of individual and cognitive structures, as exemplified by an excessive focus on personality traits [9], thrill-seeking [10,11] and risk-taking [12,13] tendencies to explain behavior, decision making, or the performance regulation of individuals. This somewhat narrow focus has led to definitions and views whereby participation is often viewed as somewhat pathological and a platform for taking socially unacceptable risks. However, a growing body of research has revealed numerous physical, psychological, and social health and wellbeing benefits of participating in adventure sports [4,14]. In this paper, we build an argument suggesting that the standpoint of conceptualizing adventure sports as solely dangerous, pathological, highly risky, unhealthy, or practiced by individuals with deviant personalities, is based on over-simplified conclusions about putative motives and psychological dispositions of participants that stem from a weak theoretical approach and reductionist research paradigm. This argument raises a critical question: How can adventure sports be better understood and conceptualized to enhance the impact of research and support practical implications?

In this article, we will briefly summarize previous definitions to address the evident confusion and lack of conceptual clarity in the discourse. It is important to understand the historic use and development of contemporary definitions and to comprehend how these definitions might have led to over-simplified research paradigms and subsequent conclusions. To clarify, in the scope of this article, we have not included an exhausting systematic review of adventure sports literature but have rather focused on ecological literature in an attempt to avoid the trap of emphasizing ‘organismic asymmetry’ [7,8]. Drawing on *ecological dynamics* as a transdisciplinary approach and holistic framework, we propose that a perspective on human behaviors as interactions with the environment may provide an appropriate conceptualization to enhance future research and practice. Ecological dynamics has been utilized extensively in traditional sport performance research [5]. This comprehensive and multifaceted ecological understanding is crucial for providing sound, evidence-based practical implications, captured in investigations of participation motivation, coaching practices, or educational structures. We emphasize the need to study the constantly evolving and distinguishable adventure sport ecological niches, captured as ‘forms of life’ [15] within and across disciplines. Such ecological investigations could reveal the incredible richness of highly individualized participation styles and personal philosophies. We believe that adopting this participant–environment scale of analysis has the potential to substantially improve and deepen our understanding of various aspects in adventure sports. Through this interpretative lens, practical implications can be utilized more broadly, or targeted more accurately.

## 2. Previous Definitions of ‘Non-Traditional’ Sports

When defining sports outside of the traditional, competitive sporting domains, terms such as action [4,16,17], adventure [4,18,19], extreme [3,6], free [20], lifestyle [21], alternative [22] high-risk [23,24] or (adventurous) nature sports [25,26] have often been used interchangeably in previous research. Evidence reveals that some definitions (such as ‘extreme’ or ‘high-risk’) are regularly rejected by many participants as non-representative of their personal experiences and typical ways of acting. Others, such as ‘action’ and ‘adventure’ sports, are also self-referenced and agreed [2,4]. In academic discourse, however, a *participant perspective* on lived experiences in adventure sports has been typically ignored when defining or conceptualizing motivations for participation. It is possible that the endeavor to find another umbrella-term to capture all forms of non-competitive and non-regulated sports, in different situations and contexts, has the potential to act as another pitfall by adding to further confusion in the already disoriented discourse. For this reason, a brief look at some previous definitions, and challenging some previous interpretations and evaluations, is necessary to create a starting point for the rationale of a key unresolved question: *What* constitutes an adventure sport?

Here, we summarize some notions from previous literature to ground the point of departure for a clearer definition of adventure sport. First, in contemporary research, activities requiring a high level of self-knowledge, personal skills, training, commitment, environmental knowledge, and task knowledge, such as big mountain snowboarding or skiing, are consistently confused with activities that require no previous experience or knowledge of the activity or environment, such as commodified white-water rafting or bungee jumping or, in some cases, even with traditional sports such as triathlon [27]. The difference between the two types of activities is framed by more and less opportunity for participant *self-regulation* (relying on perception, action, cognition to negotiate the environment in the form of problem solving and decision making). Findings from studies (for instance, on motivations or risk-perceptions) of individual participants in instructor-led, commodified activities, may not be generalizable to understanding participation in activities such as high-altitude mountaineering, off-piste (backcountry) ski journeys or self-regulated sea kayaking expeditions. Differences in self-regulation and decision-making opportunities by individuals is an important notion to consider when recruiting research participants from several sports or activity categories into the same study of adventure sport experiences.

Second, a wide spectrum of (positive) outcomes and motives for participation has been reported by participants (see summary, Table 1). This evident diversity of effects indicates that attempts to describe all forms and participation styles of adventure sports, under the umbrella definitions of ‘high-risk’ or ‘lifestyle’ sports, are fundamentally misleading. In addition, as this diversity exemplifies, outcomes cannot be understood solely as pathological or unhealthy for participants. 

Third, sports differ in terms of activity duration and intensity, and it is important to note that this distinction can lead to different interaction effects on behavior or experiences. For example, an expedition to Everest might take weeks (and months of planning), which exposes individuals to prolonged periods of environmental, social, and psychological uncertainty, whereas the performance window in some organizational- or instructor-led activities might be only a few seconds or minutes. This notion sets specific requirements for methodological approaches and eligibility criteria of research participants in studies.

Fourth, in contrast to previous assumptions, participants represent a broad demographic, including males and females of various age ranges and education and income levels [40], suggesting that the characterizations of groupings for data interpretation, such as ‘youth sports’, need to be seriously reconsidered as encompassing descriptions.

### 2.1. Delineations of ‘Action and Adventure Sports’ and Traditional, Competitive Sports

Many traditional sports have their roots in either religious or mythological backgrounds (e.g., ancient Olympic games) or in competitive aspirations of activities originally rooted in cultural ways of movement (e.g., the javelin throw evolved from everyday use of spears in hunting and warfare), whilst others are founded on forms of natural human locomotion, exemplified by the birth of the modern marathon run. One distinctive feature in the birth and historical developmental trajectory of *action and adventure sports* is that their origins can be traced back to recreational activities, instead of competitive or any other externally oriented aspirations. Examples include the long history of surfing with roots in the Polynesian islands [41] or snowboarding in the remote villages of Turkey’s Kackar mountains. The emphasis on the intrinsic value of the activity itself and non-competitive orientation of motivations and aspirations of participants, was descriptively and eloquently captured by early pioneers of rock climbing, who considered themselves as ‘conquistadors of the useless’ [42,43]. In traditional sports, the formal organization of agreed rules and universal statutes have replaced an informal acceptance of unwritten and conventionally fluid participant norms, such as climbing ethics or surfing etiquette, which are constrained by localized, sociocultural constraints [4].

Formally structured, universal competition characteristics do not seem to fit the sociocultural ‘forms of life’ in most action and adventure sports. Collins and Carson [2] use five rigid conditions of the *système sportif* [44], to exemplify the traditional definition of sports. These include the following: (1) a series of universally accepted applied rules and regulations codified in a rulebook, (2) the application of said rules by institutions who oversee the application of the rules to ensure equality of performance opportunity for all within the regulated framework, (3) the principle of equality of competitive opportunity to ensure a level playing field among participants, (4) a particular sporting space to be created, defined clearly in the designated rulebook, and (5), specific time durations (such as standardized periods of 3 × 20 min in ice hockey), stated in advance and laid out in the above rulebook. Although the term ‘sport’ often refers to organized structured competition, the etymological background of the term also describes it as a pastime or recreation. Thus, ‘sport’ can be considered as multifaceted, and in many cases boundary-crossing activities, which do not necessarily involve formally organized and structured competitions organized by a governing body, rules, institutions, or regulated performance environments [4].

One important, and sometimes confusing, aspect in action and adventure sports is that due to their evolution from non-competitive origins towards a variety of competitive formats, a specific sport can nowadays be seen from multiple angles and explanatory frameworks. For example, Olympic snowboarding could be defined as a traditional sport, if viewed strictly through a criterion of the *système sportif*. This is due to the fact that regulated Olympic disciplines such as half-pipe, big air and snowboard cross have clearly defined rules, performance environments, competition formats and they involve athletes attending training programs within traditional organizational structures. However, this is only one side of the sport and does not fully represent the diversity of participation styles and philosophies practiced, nor does it acknowledge the variety of complex ways of participation when individuals choose more than one specific way to participate. For instance, some individuals might choose to attend to only competitive forms and environments such as snowboard cross, or may only perform within some other specific niche, such as splitboarding in natural mountain environments. However, an Olympic freestyle snowboarder might also spend a major part of their season outside of competitions by freeriding in the backcountry (off-piste) or street-snowboarding in urban environments. Therefore, delineations of participation styles and subdisciplines within and across individuals are not always inclusive, clear or helpful. Although competitively transformed forms of these sports might capture public attention, it is important to notice that non-organized ways of participation have remained strong throughout this evolution and most participants practice their sport *outside of* competitive structures and away from governing bodies, including professionals. The shift towards traditional, regulated, and organized structures and environments has initiated the development of many evolving, distinguishing (and somewhat opposed to a competitive and high-performance ethos) niches on local and global scales, such as the snowsurfing, freeriding and splitboarding movements in snowboarding. This diversity and constant, unregulated evolution of formats of play and performance in action and adventure sports is now infiltrating more traditional sports such as basketball and football, producing ‘street’ and ‘cage’ versions of these sports.

### 2.2. ‘Adventure’ within Action and Adventure Sports

*Action and adventure sports* can be defined as “constantly evolving forms of activities, which, unlike traditional sports, mainly evolve and develop into their distinct disciplines without the influence of organizational structures, codified rules or clearly defined and regulated performance environments”, and “activities which flourish through creative exploration of novel movement experiences, continuously expanding and evolving beyond predetermined environmental, physical, psychological or sociocultural boundaries” [4].

Some scholars have proposed that the (natural) environment has an inherent defining role in *adventure sports* [18,25]. In the same vein, Collins and Carson [2] suggest that the (lack of) a formalized regulatory focus should be considered as one crucial defining aspect. They proposed a continuum to look at activities more specifically through an environmental focus, in which performance environments, with different participation styles, can be located (see Figure 1).

Exemplified again in snowboarding, Olympic half-pipe takes place in a *wholly manufactured* environment, specifically made for this purpose and providing conditions and clearly defined physical boundaries for performance. A *managed* performance environment is pre-built for other purposes, such as stairs and locomotion-supportive handrails, which *afford* performance engagement for urban street snowboarders. A *Modified* environment is a physically altered, natural environment, such as a ski resort with a lift system, providing a traditional environment for most recreational snowboarders. A *maintained* environment is natural, but some degree of human intervention is involved to ensure safety, such as in freeride competitions, wherein a mountain face provides an arena for performances and snow conditions remain mostly natural, but avalanche hazard is controlled. A *Natural* environment is exemplified by uncontrolled mountain conditions in the backcountry, where big mountain snowboarding or splitboarding takes place in the wilderness, outside of ski resorts.

This continuum illustrates how the typical environment in adventure sports is in a state of constant change (of natural conditions) and, according to our conceptualization proposed here, can be seen as located towards the natural side of the spectrum presented above. That is, the most important defining feature of an adventure sport is that the environment remains unconstrained (as contrasted to sports with very clearly defined and demarcated boundaries, such as a net and marked lines in tennis). This means that participants need to be highly attuned to varying information within discipline-specific, natural environments, such as changing weather and snow stability in backcountry skiing, to adapt their actions by making effective decisions, self-regulating to achieve the specific performance goals they set, or even just by avoiding injury [3]. On the other hand, for participation styles where the focus is placed more on the task or on the ‘action’ itself, the environment is often modified to support the development of action in a specific direction. This approach is exemplified by artificial climbing walls, which support high physical performance while facilitating a level competitive field in lead- and speed-climbing disciplines. Another example concerns the expression of unique, individualized styles of movement, afforded by supplementing features of an urban environment by building snowy take-offs and inclined landings to perform specific tricks in street snowboarding [4]. This approach emphasizes the importance of different manifestations of a particular sport (for instance, within and across climbing or snowboarding subdisciplines), which can have fundamentally different physical, psychological and sociocultural constraints.

### 2.3. Characteristics of Extreme Adventure Sports

Depending on research questions or practical contexts, it might be useful, and is sometimes necessary, to recognize the most ‘extreme’ participation styles of adventure sports as their own, distinct activity category [3]. This is especially important when investigating specific psychological, existential, or sociocultural variables in these activities. Whilst extreme sports have evolved from similar foundations and share mostly common characteristics with the family of action and adventure sports, they are, fundamentally, a different category of activities with distinct equipment, skills, psychological characteristics and so on. For example, snowboarding or skiing on ‘green’ beginner slopes is not an extreme form of these disciplines. Whereas a big-mountain expedition in the Himalayas, including steep descents >50 inclines, may be more extreme, according to our definition proposed here.

However, it is crucially important to understand that the experience continuum of beginner to expert is not synonymous with non-extreme to extreme participation. Participants’ developmental trajectories are individualized, and inherently multidimensional and some highly skilled expert athletes never participate at an extreme level [3]. From psychological and existential points of view, the difference between extreme and non-extreme sports is the exquisite, emerging experiences achieved during these specific participation styles, and the ensuing changes in ways individuals explore, experience, and perceive the properties of the environment, their everyday life, and fundamental human values [3,6,38]. Research has shown that the profound person–environment relationship developed via a participation in extreme sports can act as a facilitator to a deep, positive understanding of *self* and its place in relation to specific properties of the environment [45,46]. For example, experiencing extreme elation or intense fear can be a potentially meaningful and constructive event in the lives of participants, having implications as a potentially developmental and transformative process [47]. Importantly, these behavioral experiences can only occur when specific performances, often by facing danger, injury, or potential death, make deep existential structures visible and available to be experienced. According to insights from contemporary phenomenological research, these experiences are not as readily available within traditional, highly regulated sports or other, non-extreme participation styles within action and adventure sports. Adopting a phenomenological account and an ecological dynamics rationale, extreme sports can thus be defined as “emergent forms of action and adventure activities, consisting of an inimitable person-environment relationship with exquisite affordances for ultimate perception and movement experiences, leading to existential reflection and self-actualization as framed by the human form of life” [3]. This kind of demarcation between activity categories might be useful in, for example, research, when considering eligibility criteria of sports and (sub)disciplines from which to include participants. For instance, BASE or Free solo climbing might align well together due to their similar historical, psychological, or sociocultural distinctions within their own specific sociocultural frames of references, whereas bungee or drag racing are probably representative of different characteristics. To summarize, adventure sports, in both extreme and non-extreme ways of participation clearly require specific skills, personal devotion and commitment to developing necessary skills. This is not the case when individuals participate in commodified activities such as bungee jumping, where preparation and specific skills are unnecessary. Understanding these nuances in sport performance definitions is crucial in recognizing, for instance, how a variety of sociocultural values have constrained emerging participation styles [4].

## 3. Constraints, Affordances, and Form of Life in the Adventure Sport Context

Underpinning previous psychological research of human behavior in uncertain adventure sport performance environments and information-processing approaches in the study of human movement, is the dualistic premise that the individual and their surrounding environment are fundamentally considered as two separate systems. This perspective stands in stark contrast with the ecological approach to human behavior. In this section, we explain how our understanding of adventure sports can be enhanced by considering *the ecological dynamics* framework. Ecological dynamics conceptualizes humans as dynamic, complex systems constantly interacting with other systems [5,48]. In other words, as opposed to traditional approaches, the individual and their social, physical, and cultural environments are fundamentally seen as intrinsically and deeply linked, nested systems, whereby the behaviors of individuals self-organize over time under interacting *constraints* [3,49,50]. Rather than being imposed by a pre-existing, inherent structure such as an ‘action plan’, motor program, specific personality trait or an individual’s inherent risk-taking tendency, behavior *emerges* from this confluence of interacting constraints. Inherent tendencies and environmental and task constraints continually interact to provide the boundary conditions over different timescales that shape emerging behaviors on an individual’s path towards achieving specific tasks or goals [5]. In an ecological dynamics rationale, perceptions, cognitions, and actions are conceptualized as self-organized, interacting phenomena, emerging from the continuously dynamic interplay of a performer’s action capabilities (*effectivities*) and opportunities for action (*affordances*) [51] available in a specific performance environment (*ecological niche*) [52,53,54].

### 3.1. Constraints

*Constraints* are boundaries or features that shape the emergence of each individuals’ cognitions, perception, actions, and decision-making processes (see Figure 2) [55]. The three main categories of constraints are as follows: *individual* constraints, which can be structural (e.g., height, weight, body shape, technical abilities, connectivity of synapses in the brain), *functional* (e.g., motivations, attitudes, emotions, cognitions, perceptions, metacognitive capacities) and *experiential* (e.g., development to tolerate lack of comfort, learning through past experiences of accidents or “close calls”) [5]. Individuals are active agents with different personal characteristics or tendencies which may shape the distinct strategies used to coordinate actions and solve problems [56] in uncertain environments.

*Task constraints* include specific rules associated with activities, including task goals, objects, equipment, surfaces, boundary markings such as trails, signs and environmental features [57]. Adventure sport often involves the participant interfacing with a challenging environment using technology and equipment in their interactions, exemplified in backcountry snowboarding and skiing, canoeing and mountaineering. Many of these features might fit well with the definitions of traditional, regulated, organized and competitive sports. However, this is a crucial difference in comparison to adventure sports, since they are predominantly free of organizational frameworks, regulated competitive structures or rule-bound task constraints [3,4]. Indeed, freedom from regulatory tasks and environmental constraints can be seen as the most fundamental definitive characteristic of an adventure sport [2,4].

*Environmental constraints* can be physical (e.g., weather, ambient light, temperature, gravity or for instance wave interval, shape and size of swells in surfing), or sociocultural (e.g., values, family expectations, peer support, (sub)cultural norms or expectations) [4,5]. In adventure sports, a definitive feature is that physical constraints are not restricted by predetermined environmental boundaries (such as courts or arenas in invasion sports), but can involve surrounding natural conditions, such as characteristics of mountain terrain, river rapids, a track or trail, weather, visibility and snowpack features and stability in back-country skiing. Thus, delineations between traditional sports and adventure sports go well beyond the competitive vs. non-competitive dichotomy [2]. From the ecological point of view, the inherent uncertainty and sheer variability of natural adventure sport environments (as opposed to stable manicured environments, such as indoor arenas in tennis and gymnastics), provide an innate instability and dynamicity of information sources that surround the performer at all times, available for use to functionally adapt and regulate their actions and behaviors.

### 3.2. Affordances

One of the fundamental ideas of ecological dynamics is the theory of *affordances*, originating in the work of James Gibson. The concept of *affordance* refers to opportunities or invitations for actions that emerge as individuals interact with critical information from the environment [58,59]. For instance, different surfaces, substances, events, objects, or other individuals in the environment can *afford* different possibilities for actions in different people relative to their individual capacities, needs, values and motivations, acting as constraints. *Effectivities* are complementary action capabilities that can help each individual realize affordances in coherent forms of behavior [60], i.e., the dispositions, tendencies, skills, capacities and capabilities an individual can possess within a specific form of life [61]. Such coherent behavior is exemplified by a skilled freeride snowboarder (attuned to a specific *field of affordances* in a snowy environment through experience and learning). An individual might choose a steeper and more exposed line to snowboard down the mountain face, controlling the speed with line choices of carving turns instead of slowing the speed down with skidded turns. Conversely, they may prefer organizing a more technical action when jumping off a cliff compared to a novice (for whom the same *landscape of affordances* is available, but with different effectivities) [62]. Importantly, effectivities might be limited or enabled by environmental constraints, such as the natural characteristics of a performance environment or by values, social habits and attitudes [60]. Thus, information can be perceived as relational and influenced by the specific intentions of each individual and their opportunities and capabilities to interact with the environment [63]. In this way, affordances are the starting point for the ecological study of what humans perceive, what they learn and know, and how they decide and act [52,64]. When the description of the environment is founded in affordances, it changes the description from physical to functional, emphasizing the opportunities for interaction. Indeed, the environment is described in terms of what it offers a performer, ‘for good or ill’ [51]. In ecological psychology, this idea signifies that, when perceiving possibilities for action (such as safe or unsafe passages of travel in the mountains, rivers or trails), one would *directly* perceive their ‘goodness’ or ‘badness’ of fit in relation to one’s skills, values, needs, intentions, motives, emotions, interests, and goals.

### 3.3. Adventure Sport Niche as a Form of Life

Wittgenstein [15] proposed the concept *form of life*, which refers to the potential and common behaviors available to a specific group of organisms (such as group of humans or a species of birds), influencing how the group interacts with and within the world around them [61,62]. For humans, effectivities are not only relative to a particular individual perceiving or detecting affordances, but they have an existence relative to a set of skills and capacities available in a specific practice context, such as within a particular adventure sport niche. A form of life thus implies that sociocultural practices (constituted by skills, values, beliefs, habits, customs, attitudes and so forth) of humans constrain the emergence of specific behavioral patterns [15,65], such as preferred or established ways of acting physically, intentions, heuristics (e.g., ‘rules of thumb’ in complex decision-making) or attitudes towards risk-taking within the sociocultural frame of reference. The influence of affordances on the evolution and formation of niches was depicted early in Gibson’s original definition:


*“Ecologists have the concept of a niche. A species of animal is said to utilize or occupy a certain niche in the environment. This is not quite the same as the habitat of the species; a niche refers more to how an animal lives than to where it lives. I suggest that a niche is a set of affordances.”*
[51] (p. 128)

Thus, for Gibson, the concept of an ecological niche referred to a set of behaviors, capacities and characteristics required to actively engage with the surrounding information, available in energy arrays of the environment. When individuals perceive affordances of an ecological niche as feasible possibilities for actions, they will effectively orient to, and potentially start regulating their behaviors in relation to the situationally salient affordances [59,66,67].

Foregrounding the role of sociocultural constraints on behaviors, the individual-environment system can be characterized as an ecological niche that arises from, and mutually co-creates, a form of life [3,49]. In this line of thinking, each individual-environment system co-exists as an open, dynamic system, meaning it is nested within, and capable of exchanging energy and information with the surrounding ecology at micro and macro scales [49,68]. The nestedness of affordances is understood as multiple affordances that exist in any given situation, and the consequential possibilities for humans to choose among them. Given that affordances reflect the relational nature of multiple properties of individuals as well as the multiple properties of (physical, social and cultural) environments, affordances are considered to be nested in the context of other affordances. This idea implies that an individual affordance may be superordinate or subordinate to other affordances [69,70]. For example, a mountaineer ascending a mountain face might be able to find a route up towards the summit (i.e., subordinate affordance), but may fail to pass the ‘crux’, such as a steep and unstable icefall (i.e., superordinate affordance), as they may not have the required skills to do so (within their form of life). The notion of affordances offers us a valuable perspective, especially when examining performance or learning over different timescales, as it changes the perspective from a strictly positivist lens (e.g., biomechanical or physiological), towards a more relational focus of attunement to information available in the psycho-socio-material environment. This notion specifically places the spotlight on the uncertainty of typical environments in adventure sports. From an ecological perspective, expertise in adventure sport can be understood as a skilled engagement with the specific arrays of available environmental information. This is an important notion from the point of view of practical implications since training outside of a specific context (such as in a stable practice environment that is non-representative of unstable informational properties of a performance environment in nature) has been shown to be ineffective from the perspective of the development of skilled actions in multiple traditional sporting domains [48]. In other words, context is everything in sport performance. For instance, learning to swim in a swimming pool might not prepare an individual in terms of their water competence sufficiently for negotiating the affordances of different aquatic environments available in a river, lake or ocean surf [71].

Deriving from a study on football from Vaughan et al. [49], the dominant form of life (specific way of doing things) in (national, regional, or trending) an adventure sport niche might be conceptualized as deeply acculturated, socially accepted, and often taken for granted. This idea is exemplified in the frequent use of helmets and other safety gear or a social push against or towards attending to structured avalanche education in backcountry skiing and snowboarding communities. It is thus proposed that ways of acting as individuals and in groups are sociocultural artifacts [72], embodying the manifestation of the relational environment. Research studies have often included participants from distinct sporting domains or unique activity categories (e.g., bungee, and mountaineering participants as a collective, assumed to represent a population of extreme sport participants). This blanket and unrepresentative sampling approach exemplifies the tendency of previous research to ignore the role of sociocultural constraints as an important defining feature of participation style. It fails to understand the importance of the nuanced experiences and actions of participants emerging from the interaction of constraints in specific situations and contexts. Thus, researchers and practitioners, trained within the traditional positivist, reductionist research paradigms, might gain an ontologically limited picture of the complexity of human behavior, experiences, and development of skills and expertise [49]. This is arguably a serious limitation on the evaluation and interpretation of data, syntheses of study findings and general study designs.

### 3.4. Skill as an Essential Element in the Formation of an Adventure Sport Niche

Tim Ingold’s [65] views on the embodied skills of humans as fundamental attributes of cultural variation can help us to distinguish several useful levels of analysis when examining adventure sports, drawing demarcations between niches. The work of Woods and colleagues in sport science has been to develop an understanding of the importance of these ideas [73,74]. 

Three levels of analysis are exemplified in this article through the evolution of niches with a basis in an environment which affords gliding and sliding possibilities for people with an adequate behavioral repertoire [75]. These are:(1).Human forms of life have general action and movement capabilities due to phylogenetic and ontogenetic development processes over millions of years (contrasting with development of, for instance, birds or fish). Additionally, there has been the potential to collectively manufacture and utilize equipment and tools, which afford gliding on different surfaces in different contexts (such as snowboarding or skiing as recreation, sledging for transporting food or cross-country skiing as locomotion or locomotion-aid in hunting)(2).There exist specific and distinguishing sociocultural practices, i.e., regularities in the performances, behaviors and experiences of groups of people who utilize gliding and sliding movements recreationally or competitively in specific contexts, situations and geological locations. Examples include snowboarding in urban environments, executing rail tricks, performing carving turns on maintained runs of ski resorts, or splitboarding utilizing mountaineering skills to vertically traverse natural mountain environments.(3).A more detailed analysis indicates that skilled engagement with affordances is highly individualized, diverse, and multi-dimensional within a specific sociocultural frame of reference. For instance, an individual might choose to attend to a half-pipe competition one day to win, snowboard at a resort for personal recognition among their peers on the next day, ride an extremely steep and challenging descent on an untouched mountain face to undergo life-changing and transformative experiences on a third, and hire a mountain guide to explore a new mountain range on the fourth day.

Researchers and practitioners need to be clear on the contexts framing each kind of activity being studied and how the results of analyses may be generalized and utilized by making use of this kind of comprehensive background analysis. From an ecological perspective, the important point is that the development of skills is a fundamental component in how niches are mutually formed and evolved through person–environment interactions. Importantly, the evolution of a niche mutually affords a physical, psychological, and sociocultural environment for individuals and groups to become attuned to, and therefore, utilize available affordances and develop skills even further.

### 3.5. Adventure Sport Niche as a Dynamical Complex System

Another level of analysis in the ecological approach is that a form of life can be understood within the idea of adaptive and nested complex systems, meaning that systems that interact with other systems are simultaneously integrated with other systems. During life-long timescales of learning, the influence of multiple components of the broad web of interconnected systems on each individual’s development (e.g., towards development of skilled decision-making) becomes more complex to understand. Thus, hypotheses, for instance, on development of expertise or perceptual-motor learning, become more challenging to formulate. Here, Bronfenbrenner’s [68], bio-ecological model provides a reference framework to better focus on different levels of an inherently interconnected and dynamic complex system (see Figure 3). For example, the neuro-biological, psycho-social, and socio-cognitive subsystems of an individual freeride snowboarder or skier, perceiving and acting within the contexts of local/global freeride niches, are enclosed by the ecologies of: (i) broader ski- and snowboard cultures, (ii) action, adventure-, and extreme sport cultures, (iii) (local and global) outdoor culture(s), (iv) national and global (sub)culture(s), (v) human forms of life as a whole, (vi) local, and global safety, political, economic, geological and (vii), weather systems and so on. To date, no research has been conducted to explore the mechanisms and nature of relations among these interactions in the adventure sport context from this perspective.

### 3.6. Characterization of Skill and Skill Transfer in Adventure Sports

In many traditional sports, skilled performances can be defined and measured quantitively, e.g., in units of time, distance or score or in comparison to other participants. However, this kind of assessment makes it difficult to comprehensively understand skill or expertise in adventure sports. Specifically, the meaning of concepts such as ‘success’, ‘winning’ or ‘losing’ are challenging to externally, or quantitatively, define in adventure sport contexts. This idea can be seen as another defining aspect of adventure sports, as participants might need to figure out for themselves what success might mean in their given activity. For instance, Everest mountaineering is about getting up and back down safely, not just reaching the summit in a record time. As rules typically do not govern how to win or participate in adventure sports, the structured competition perspective does not provide a very fruitful approach to measure or define skilled or successful performance [4].

In adventure sports, the functionality of skill (i.e., how effectively task goals are achieved) partially depends on subtle interactions of task and personal constraints such as originality, collective agreement, and interpretation [4]. For instance, a climber’s attempt to climb a new route using quickdraws as ‘holds’ for pulling movements might not be recognized as ‘successful climbing’ among sport climbers. These interactions also concern the physical environment. For example, the way a kayaker decides to descend a big rapid is partially about ‘reading’ the water flows and currents and seeing what the rapid will allow (in terms of affordances), as elucidated in the social anthropological analysis of enskillment as dwelling in habitation by Woods and Davids [73]. These are examples of sociocultural tendencies within a form of life, or a ‘field of promoted actions’ [76] which can significantly constrain the behaviors of individuals or reinforce those that have been established or are preferred. Importantly, specific social mores and values (due to specific historical and social constraints) might emerge to invite participants to function, by challenging rules and norms to create their own, distinctive style of movement or unique approaches towards the specific environment. Therefore, a development of expertise in adventure sports requires a deep and contextualized understanding of the underpinning sociocultural constraints on functional practices (i.e., based on their usefulness, effectiveness, appropriateness, or adequacy), supported by the knowledge of how to diverge from them with innovation and novelty [4].

The fundamental challenge of becoming an expert (or to study in authentic contexts how to become one) in adventure sports is that a typical performance environment is naturally challenging, uncertain and often hazardous, meaning that there are not always possibilities to perform and practice (or run experimental tasks) safely in that environment. For example, the constantly changing avalanche hazard in a mountain environment affords no serious mistakes during the learning of skiing locomotion in avalanche-prone territory, due to the obvious reality that mistakes in judgement or decision making can be fatal. This type of affordance landscape sets a unique requirement in that skills must often be learned in environments that allow for safe exploration and possibilities to learn through trial and error. Accordingly, the boundary conditions of practice and training (as well as experimental tasks) need to allow for these specific skills to be functionally adapted for use in, or ‘transferred’ to, natural performance environments.

### 3.7. The Relations between Generality and Specificity of Practice and Learning in Adventure Sports

One important aspect in understanding skilled adaptations to new performance environments (task requirements) is that the greater the similarity between the individual’s existing coordination tendencies and those required in a performance environment, the more likely skill adaptation will be functionally relevant. This is mainly due to the increasing functional alignment that can emerge between an individual’s intrinsic dynamics (effectivities/behavioral repertoire) and specific task dynamics with practice and experience [50]. Therefore, *skill transfer* can be understood as a function of proximity, relevance and stability of coordination patterns in one’s behavioral repertoire to changing task requirements and the high capacity to attune to information in the environment to regulate actions. In this line of thinking, skill transfer emerges when intrinsic and task dynamics coincide in terms of behavioral or movement patterns [47,77,78,79]. Skilled performance in adventure sports is not limited to the performance of physical motor skills or techniques. Rather, a *‘functional behavioral repertoire’* in adventure sport also includes a discipline- and context-specific set of perceptual and cognitive skills (such as problem solving, perceptual judgment and decision-making skills when choosing safe passages of travel in avalanche terrain), i.e., a deep knowledge of oneself, interacting with the task, and specific physical and sociocultural environment [3]. Thus, the challenge is, for practitioners such as psychologists, coaches, instructors, and applied scientists working within adventure sports, to develop learning contexts which effectively prepare participants for negotiating the multitude of demands and the uncertainty of future performance, *wayfinding* to self-regulate [80].

*General skills adaptation* has been defined as the capacity to explore information for action in the environment. That is, when the initial intrinsic dynamics of the individual are not closely aligned to the expected task dynamics, the performer can use general foundational movements and capacities that exist in their repertoire (e.g., thinking, problem solving, decision making, coordinating actions, anticipatory skills or visual search strategies) to continually satisfy emerging task constraints. *Specific skills transfer* emerges under precise practice conditions and is understood as the capacity to perceive and utilize specific affordances or to use an existing specific coordination pattern in a new task [81]. Notably, general transfer can support specific transfer [82,83]. In adventure sports, general and specific transfer have been cogently evidenced and exemplified in the climbing context. For instance, the studies of Seifert and colleagues revealed how specific, positive transfer occurs in the case of intermediate rock climbers’ coordination dynamics when transferring to an ice-climbing environment. Specifically, an exploration of relevant information and affordance-perception in ice climbing has been shown to be positively affected by previous rock-climbing experience, exemplifying the occurrence of general transfer between indoor and outdoor environment [81,82,83,84].

Results from studies utilizing the framework of the Athletic Skills Model [85], have revealed transfer processes in traditional sports, suggesting that engaging in a ‘donor sport’ [86,87] can effectively aid the acquisition of motor skills and the development of expertise in another closely-related sport or activity context, provided that the donor sport is of a complementary nature to the target activity to ensure positive skill transfer. An example of how donor sports in adventure sports may induce positive transfer exists in the case of a backcountry snowboarder, with a long history of balancing on boards in surfing, skiing and skateboarding, developing fundamental motor skills in parkour and learning to ‘read’ mountain terrains in alpine climbing. The activities performed in donor sports provide general experiences in tasks that demand balancing and changing direction while rolling, gliding and sliding when locomoting on different surfaces such as tarmac, waves, snow and ice. The relationship between generality and the specificity of learning and practice and their effects on sport performance is a hot topic for future research [88]. As well as general learning experiences, an ecological dynamics rationale emphasizes the specificity of the learning principle in the concept of representative learning design [89].

### 3.8. Representative Design as Precursor of Skill Acquisition and Expertise in Uncertain Adventure Sport Environments

Brunswik [90] proposed the concept of *representative design* to refer to conditions and information in psychology experiments. It emphasizes that because participants of an experiment need to be precisely representative of those to which the study wishes to generalize, experimental task constraints must rigorously represent the behavioral constraints to which they are to be generalized [89,90]. This idea of representativeness is a particular concern for the study of human behavior in uncertain environments and adventure sport contexts. From an applied point of view, if experimental tasks do not account for representativeness of a performance context (e.g., experiments executed in indoor laboratory settings, or via video, internet, etc. instead of authentic performance environments), they may not support a useful analysis of the critical aspects of the skills required, or to be trained. Nor will they allow for any further development of intervention or training tasks in order to achieve these aims [91].

Pinder and colleagues [91] applied Brunswik’s idea of representativeness to the design of practice and learning environments in sport. In ecological dynamics, this premise is described by the concept of *representative learning design* [89]. It emphasizes that for principles of representative design to be applied to the design of functional interventions, practice, and training tasks, it is crucial to acknowledge that different sources of perceptual information present different affordances for different individuals. In other words, how adequately the constraints of practice tasks replicate the specific context (i.e., the performance environment) is of great relevance so as to allow participants to detect affordances for action and couple actions to key information sources within those specific settings [91].

Recently, Woods and colleagues [52] proposed a concept of *representative co-design* to better understand and utilize experiential knowledge of experienced participants in enriching the designs of learning environments. This idea suggests that, with increasing experience and expertise in sport, performers evolve in terms of their decision-making by becoming increasingly competent at realizing the most soliciting or inviting affordances within their ecological niche [52,58,59]. Representative co-design is predicated on Gibson’s distinguishing ideas on *knowledge of*, and *knowledge about* the environment. *Knowledge of* the environment refers to understanding of the use of affordances in regulating interactions in a performance environment, analogous to a ‘first person point-of-view” (relation of an individual athlete’s unique intrinsic dynamics and environment and task constraints). *Knowledge about* the environment, on the other hand, facilitates an internalized symbolic manifestation of the environment available (metaphorically, a point-of-view of a sports commentator observing and analyzing performance in climbing competition). Importantly, it might be especially useful for practitioners and researchers to utilize this idea and emphasize this crucial distinction by exploring experiential knowledge and (co-)designing, with experienced and skilled participants, representative learning activities that specifically develop each participant’s *knowledge of* a performance environment [51].

### 3.9. How Ecological Dynamics Can Enhance Research and Practice in the Exemplary Field of Avalanche Education and Research

A current hot topic within the adventure sport research, exemplifying and underlining the challenge of context-dependence in uncertain and complex environments, concerns the behavior and decision making of backcountry skiers and snowboarders in avalanche terrain. In the field of avalanche research and education, contemporary theoretical approaches have been mainly adopted from fields such as behavioral economics, aviation, or military, alongside psychology, social, and cognitive sciences [92,93,94]. These traditional approaches have typically focused on the relations of a somewhat debatable concept of *human factors* and prevention of accidents. This lens has emphasized *human error* as one of the main factors in explaining human actions and behavior in uncertain backcountry environments, often including the assumption that humans weigh up their decisions and actions (e.g., “should I ski this slope”) in relation to negative/positive outcomes (e.g., enjoyment, pleasure or social recognition among their peers) [95]. In this line of research, causal mechanisms have been assumed to exist between flaws in decision-making processes and/or social dynamics and avalanche-related accidents. This approach is often interpreted to be due to a situational inability of individuals to perceive probabilities or environmental information indicating avalanche hazard [94]. This *cognitive bias and heuristics paradigm* is exemplified in the concept of *‘heuristic traps’* [92,96], which is widely applied in avalanche education curricula, for instance, as a basis of decision-aids or checklists as practical tools to reduce error [93,97]. Despite the width of application, empirical evidence on the effectiveness of education on heuristic traps is not yet available in the peer reviewed literature. Evidently, the prevention of accidents should be an important priority. In addition, the past use of human error terms, theories, and methods has unquestionably led to progress being made in the field. However, acknowledging the complexity of uncertain (physical, social, and cultural) environments and interacting constraints on behavior, these approaches have obvious and significant limitations as a basis for a comprehensive and nuanced understanding or as a broader theory of human decision making in avalanche terrain.

Few studies have analyzed behavior in avalanche terrain beyond the traditional risk-based approach. Some attempts have been made to include the influence of specific contexts to analyses and research designs (see [97] for a recent example). Contemporary disciplinary approaches are based on the premises of the ‘covering-law model’, invoking universal and general statements to explain patterns in human behavior [98,99]. Arguably, this kind of approach fails to account for the complexity of individualized behavior and the development of skilled actions over varying spatiotemporal scales, embedded in a multitude of sociocultural contexts and interacting systems. Importantly, in the field of *human factors and ergonomics* (outside of avalanche research), the use of human error terms has shifted towards a broader view of ‘system level failure’, and has come to be better understood and explained through theories and concepts of non-linear complex systems (for a recent review, see [100]). These examples indicate that there is a pressing need for more research, which recognizes the study of human behavior in uncertain avalanche terrain as a multifaceted *wicked problem*, requiring transdisciplinary approaches [49].

To understand the multitude of sociocultural constraints that influence situational behaviors (and importantly, learning and development over longer timescales) of snowboarders, skiers, snowshoers, snowmobilers and avalanche professionals, the notion of form of life, consisting of values, beliefs, practices and customs that continually shape how we live, provides an alternative and comprehensive approach. This idea is also rooted in the *skilled intentionality framework*, a conceptual framework that directly couples forms of life to the relevant fields of affordances, influencing skilled action [62,63,98]. Specifically, this view emphasizes the notion of *socio-material entanglement*, stressing that affordances are entwined within a more culturally encompassing, socially and historically developed constellation of practices and forms of life [63,98]. Importantly, the skilled intentionality framework can illustrate the extent to which sociocultural and historical constraints in a form of life (e.g., within a group, regional freeriding community or organization of avalanche professionals) shape the intentions of humans, soliciting some affordances over others and, accordingly, direct learning and development in skilled decision making [98].

This exemplary analysis is not an assault on disciplinary research, or a call for researchers to entirely abandon accident investigations or examinations based on traditional experimental approaches. However, it addresses the need to transcend the limitations of traditional biases in order to aid researchers and practitioners faced with context-dependent, real-world wicked problems and an inherent psychological and sociocultural complexity of adventure sport participation as exemplified in this article. Therefore, an important consideration for researchers (in addition to richness in data and methods) is that *requisite variety* (the need of tool or instrument to be at least as complex, flexible and multifaceted as the concept, interaction, situation or actors being studied) needs be theoretically supported [101]. Thus, effective theoretical frameworks for future research need to be able to include and utilize experiential knowledge from all levels of the system (i.e., avalanche professionals, recreationists, researchers, and avalanche educators), and consequently, to be able to draw necessary demarcations between and within these forms of life (i.e., distinct ‘decision-making cultures’). This is the promise of trans-disciplinarity embedded in the ecological approach, including the premise that (in contrast to mono-, multi-, and interdisciplinary approaches) trans-disciplinary inquiry inherently includes a reciprocal top-down, bottom-up dialectic between academics, practitioners, and participants [49].

## 4. Concluding Remarks

### 4.1. Understanding and Defining Adventure Sports

In this article, we have proposed how an ecological dynamics framework can help us to consider two essential unresolved questions: What constitutes an adventure sport and how can they be better understood for research and practice? We have argued that through an ecological interpretation and explanatory framework, it is possible to achieve a nuanced perspective with more detailed definitions of (i) activity categories (such as adventure and extreme sport niches), (ii) characterizations of specific activities (such as sport climbing and trad climbing) understood as specific forms of life, and (iii), perceptions, cognitions and actions of individuals within these specific activities.

Through the notion of form of life and insights from phenomenological accounts, adventure sports can be understood as ‘worlds’, analogous to domains such as ‘worlds’ of science, music, or art, which can offer a multiplicity of ways for individuals to experience and perceive the socio-material world in uniquely specific and meaningful ways. This notion expands the perspective to broadly understand each adventure sport, not solely as a recreation, pastime or a ‘sport’, in a traditional sense, but as a type of form of life specific to being human [3]. Therefore, adventure sports can act as a medium for humans to engage with the world, to experiment with one’s physical or psychological capacities and ultimately, to explore what it inherently and fundamentally means to be a human. This perspective also conveys why adventure sports should be fundamental across all human experiences and embedded into systems such as education, health, environment studies and psychological interventions. United Nations has identified human health as a key indicator of international and national development. This is an important point to consider since sustainable development goals might not be met by merely trying to increase participation in traditional sports and governance structures. Therefore, a broader application of what constitutes a sport in general can provide a more holistic response to the growing demands of sustainability in sports in the future [102].

### 4.2. Limitations

The broader discussion about the incompatibilities of the ecological dynamics framework and information-processing approaches is beyond the scope of this article. Consequently, an exhaustive review of adventure sports literature, which has been conducted from the information-processing perspective, has not been included. This has been a conscious choice. We emphasize the importance of understanding that major principles of ecological dynamics are deeply connected and intertwined. Therefore, from the ecological point of view, being eclectic when selecting principles and concepts to support practical implications (for instance, teaching or coaching methods) is not beneficial or useful. That is, to subscribe to some and discard the other ideas always entails contradiction [103].

The practical implications discussed in this article are mainly derived and grounded in the *theoretical* rationale of Ecological dynamics. Therefore, it is again worth emphasizing that, as discussed earlier in this paper, ecological research on adventure sports is in its relative infancy. Hence, to enhance evidence-based practical implications in the future, there is a pressing need for more research from ecological, trans-disciplinary perspectives. 

### 4.3. Implications for Research

Understanding the principles of the ecological dynamics rationale, as presented in this article, is especially important when sampling representative research participants, when comparing data from multiple studies or when examining behaviors, motivations, cognitions, decision making or perceptions and actions of humans at an individual level in relation to specific contexts and situations. Recent examples in developing fields, such as avalanche research and education, have indicated that to enhance practical implications, theoretical approaches and prevailing paradigms need to be critically, and even radically questioned and re-evaluated. In other words, to capture the complexity of the interactions of multiple facets influencing human behavior, to advance theoretical knowledge and improve applied practice, it is vital that adventure sport research transcends mono-disciplinary approaches, and moves beyond a paradigmatic, quantitative, and often reductionist lens.

For researchers in adventure sports, it is also critically important to have a detailed grasp of historical and sociocultural constraints of the activities under examination. This is a fundamental attribute of the ecological lens, since understanding the premise of the representative nature of situational and contextual horizons of cognitions, perceptions and actions of individuals, is an essential foundation of the individual-environment scale of analysis. From the ontological and epistemological points of view, there is an evident need for a more balanced composition of research methodologies and methods to address inherent complexity (such as the phenomena of learning, skilled decision-making, or transformative human experience). Traditional, positivist baseline assumptions and research paradigms might not be, and have not been, able to fully capture the complexity of these issues, to appropriately guide the enhancement of sound, evidence-based practical implications.

### 4.4. Implications for Practice

In this article, we emphasized the interdependence of representative research designs and practical implications. From an applied point of view, to develop functional, evidence-based interventions or training tasks which effectively develop critical aspects of the skills required in each activity, they need to be based on adequately representative study designs. 

Practical tools for practitioners include the utilization of donor sports and development of representative learning designs which emphasize possibilities for participants to become attuned to critical information in the environment, allowing safe training conditions, and a positive transfer of skills to potentially dangerous ‘real life’ adventure sport environments. We have highlighted the essential role of sociocultural constraints in the development of skills in adventure sports. This includes noticing the possibilities of exploration and a utilization of the experiential knowledge of experienced participants, practitioners and researchers when developing effective learning designs. Representative co-designs of learning tasks, and environments can allow individuals to evolve in their decision making by realizing the most soliciting or inviting affordances within their ecological niche. An important notion here is that training contexts need to adequately represent ‘real life’ contexts, specifically by comprehensively paying attention, not just to the physical and psychological, but also to sociocultural task requirements.

## Figures and Tables

**Figure 1 ijerph-19-03691-f001:**
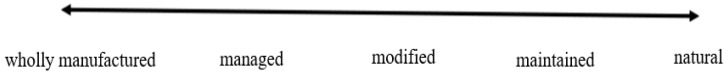
Adopted from Collins & Carson [2]. The participation context in action and adventure sports.

**Figure 2 ijerph-19-03691-f002:**
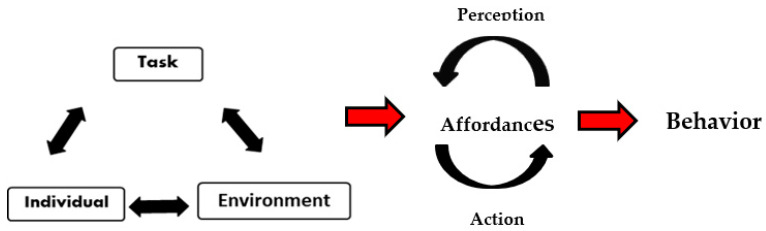
Coordination of behavior emerges from the interaction of key constraints on the performer in the form of functional information-movement couplings through system self-organization [5,55].

**Figure 3 ijerph-19-03691-f003:**
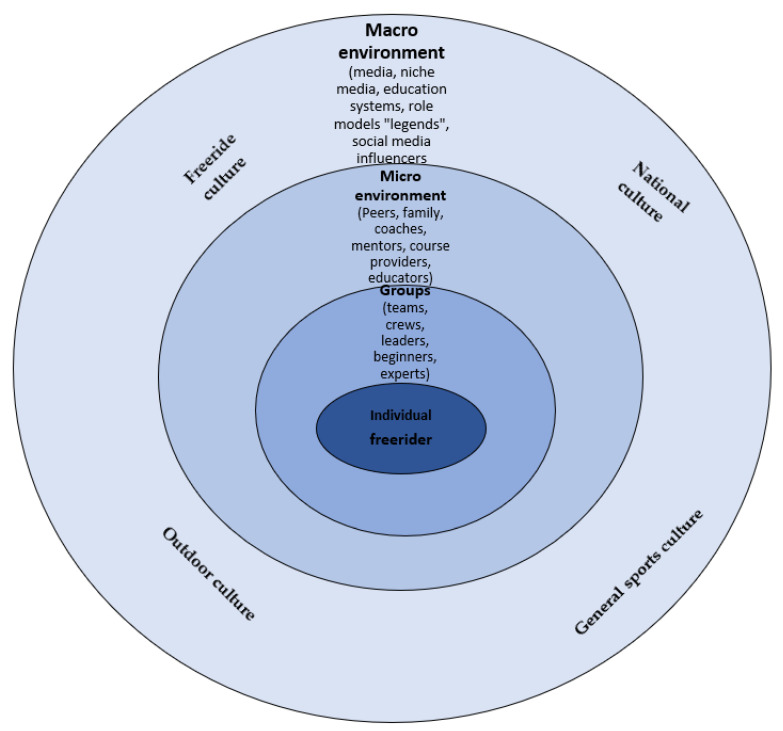
An ecological context of adventure sport participant (exemplified by a freeride snowboarder or skier).

**Table 1 ijerph-19-03691-t001:** Diversity of motives and effects of adventure sport participation.

Increased positive psychological outcomes, such as resilience and self-efficacy	(Brymer & Schweitzer, 2013 [28]; Mackenzie, Hodge, & Boyes, 2011 [29])
Experiences of connection with nature	(Brymer & Oades, 2009 [30]; Varley, 2011 [31])
Increased physical activity levels	(Clough, Mackenzie, Mallabon, & Brymer, 2016 [32])
Relieving boredom and social rela tionships	(Kerr & Mackenzie, 2012 [33])
Pushing personal boundaries and overcoming fear	(Allman, Mittelstaedt, Martin, & Goldenberg, 2009 [34]; Brymer & Oades, 2009 [30])
Enjoyable kinesthetic sensations	(Varley, 2011 [31])
Control, mastery and skill	(Allman et al., 2009 [34])
Specific goal achievement	(Willig, 2008 [35])
Contribution to deep friendships	(Frühauf, Hardy, Pfoestl, Hoellen, & Kopp, 2017 [36]; Wiersma, 2014 [37])
Overcoming challenge	(Frühauf et al., 2017 [36]; Kerr & Mackenzie, 2012 [33])
Positive transformational experiences	(Brymer & Schweitzer, 2017 [7]; Holmbom et al., 2017 [38])
Opportunities to fulfill basic psychological needs of autonomy, relatedness and competence	(Houge Mackenzie & Hodge, 2020; Houge Mackenzie, Hodge, & Filep, 2021 [39]).

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
