# Peer review of "An Ecological Dynamics Approach to Understanding Human-Environment Interactions in the Adventure Sport Context—Implications for Research and Practice"

_ijerph, 2022, doi:10.3390/ijerph19063691_

Round 1
Reviewer 1 Report
General Comments
This manuscript has the potential to be interesting as it focuses on developing a conceptual framework to improve the understanding on how the complex human-environment behaves within the adventure sport arena. I think that the quality of the text and its form are in line with the high standards of the Journal. However, I would suggest the authors to better support their claims/statements throughout the manuscript with proper citations and to explore in deep the scientific literature remaining sufficiently detached and unbiased.
I suggest more specific comments:
Specific Comments
Line 59: please add a ref
Line 70: please add a ref
Line 102: please add a ref
Lines 223-230: this occurs also within a constrained environment. A constraint itself does not avoid effective decisions or affect an individual’s self-regulation…..Try to be unbiased in this occasion.
Lines 318-328: here the authors did not take into consideration the task difficulty, which is clearly exposed in the work of Guadagnoli et al. 2004 who introduced the concept of “challenge point”. Please put effort to remain detached from your valuable opinion on the important role of ecological dynamics and its application to adventure sports. I believe that literature should be analyzed more in deep.
Lines 517-522: I recommend the authors to exercise caution at this stage. It is not all about randomness. Even a climber presents skills (specific skills) that came from not only deliberate practice within an unconstrained environment (I believe that each environment represents a constraint itself) but also from training under constrained conditions. It also applies for kayak…so on and so forth. Please clarify this point taking into consideration all the pieces of the puzzle.
Lines 559-564: again, this also applies for most of the sports. The cognitive domain is crucial for most of the sports performance.
Author Response
This manuscript has the potential to be interesting as it focuses on developing a conceptual framework to improve the understanding on how the complex human-environment behaves within the adventure sport arena. I think that the quality of the text and its form are in line with the high standards of the Journal. However, I would suggest the authors to better support their claims/statements throughout the manuscript with proper citations and to explore in deep the scientific literature remaining sufficiently detached and unbiased.
Response: We thank reviewer form constructive comments. Below, we have discussed specific comments in detail and pointed out revisions which have been made.
I suggest more specific comments:
Specific Comments
Line 59: please add a ref
Response: rephrased to clarify this sentence, see line 53.
Line 70: please add a ref
Response: reference added, see lines 68-69
Line 102: please add a ref
Response: reference added.
Lines 223-230: this occurs also within a constrained environment. A constraint itself does not avoid effective decisions or affect an individual’s self-regulation…..Try to be unbiased in this occasion.
Response: Section revised to clarify our point, see lines 223-230.
Lines 318-328: here the authors did not take into consideration the task difficulty, which is clearly exposed in the work of Guadagnoli et al. 2004 who introduced the concept of “challenge point”. Please put effort to remain detached from your valuable opinion on the important role of ecological dynamics and its application to adventure sports. I believe that literature should be analyzed more in deep.
Response: We agree that remaining unbiased is a crucial point to consider. However, since the aim of this manuscript is to build a conceptual framework to understand adventure sports specifically from the ecological dynamics perspective, exhaustive literature review including articles based on information-processing approaches has therefore not been included more broadly (see revision, lines 65-67). Also, approaches based on information-processing often exemplify ‘organismic assymetry’, the point which we have emphasized earlier in this manuscript and provided example articles to clarify this in sport science and psychology, see line 46. We acknowledge that task difficulty, skill level, and ‘optimal challenge point’ can have important roles to consider in motor learning. However, we think that wider discussion about the contrasting views of information-processing and ecological approaches is beyond the scope of this manuscript. Importantly, the exemplary paper suggested by reviewer (Guadagnoli et al., 2004) is based on the information-processing approach, which is incompatible with the ecological approach, especially in terms of the role and nature of information in the regulation of human movement and behavior (predictive control vs. prospective control, role of memory etc.). Also, as the aim of this manuscript is to build a conceptual framework based specifically on the ecological perspective, this particular section about constraints is written to exemplify the role and influence of interacting constraints (and how constraints can partially help us to define adventure sports), as understood in Carl Newell’s constraints model (1986), which is an essential element in ecological dynamics. Therefore, we thank for the suggestion and acknowledge reviewer’s point about task difficulty as an important one, but we argue that it needs to be understood fundamentally differently in ecological vs. information-processing approaches. For instance, as individual’s skills evolve, so does the availability and ‘grip’ of environmental information, perceived and used to regulate specific action(s) by such an individual. In other words, as the task difficulty increases (and thus movement / skill evolves), so does the perception and utilization of affordances of the (discipline- and context-) specific environment within a form of life. This has not only spatial, but also sociocultural and temporal dimensions. Arguments about the role of affordances and form of life are built in the next sections of the manuscript.
Minor revision done, see lines 291-292.
Also, see revision, lines 65-67.
Lines 517-522: I recommend the authors to exercise caution at this stage. It is not all about randomness. Even a climber presents skills (specific skills) that came from not only deliberate practice within an unconstrained environment (I believe that each environment represents a constraint itself) but also from training under constrained conditions. It also applies for kayak…so on and so forth. Please clarify this point taking into consideration all the pieces of the puzzle.
Response: We do acknowledge this point that it is not completely about randomness. Training in constrained and unconstrained environments is discussed in detail later in this manuscript within the different concepts of transfer and examples are provided e.g. in the context of climbing (donor sports / transfer / affordance perception between indoor and outdoor environments etc.)
Lines 559-564: again, this also applies for most of the sports. The cognitive domain is crucial for most of the sports performance.
Response: Agreed, revision done to sharpen our point here, see lines 573-576.
Reviewer 2 Report
Please check the attached file.

Author Response
The authors presented “An Ecological Dynamics Approach to Understanding Human Environment Interactions in the Adventure Sport Context – Implications for Research and Practice”. The authors should be applauded for a novelty study. However, I found that some parts can be written more clearly so general readers can understand easily. The specific comments can be found below:
Response: Many thanks for reviewer for constructive comments and feedback. Below, we have discussed specific comments in detail and emphasized revisions which have been made.
1.In the introduction, authors mentioned whether adventure sport are sports at all? I don’t think having this rhetorical question is evident enough as a reason to investigate the purpose of this study. In addition, all definitions of “adventure sport” may not share exactly similar conceptualization but they do share some conceptualization. Authors should provide more citations to support these statements.
Response: Revised as suggested
2.It seems that the purpose of the study and research question is all over the place. It is highly suggested that authors may consider traditional way in writing the purpose or research questions at the end of the introduction, so readers may be clear what this manuscript seek to investigate.
Response: We thank reviewer for this constructive comment. However, as this was not an issue raised by other reviewers, we have decided to keep the structure of introduction section unrevised and hope that the aims of the manuscript are now sufficiently clear in the introduction section (also, a rhetorical question removed as suggested)
- In the Delineations of ‘action and adventure sports’ and traditional, competitive sports section authors provided a YouTube link, which is not necessary. However, if authors want to include the link, it should be cited correctly, and authors should ensure that there is no copy right issue.
Response: Link removed as suggested
4.In the conclusions section, authors should have subheadings of practical implication and theoretical implication, since this study would like to contribute implications and research practice from the ecological dynamic approach.
Response: As suggested, subheadings “understanding and defining adventure sports”, “implications for research” and subheading and complete section of “implications for practice” have been added to the conclusions section.
5.The author provided did not provide any limitations in this study. The authors should provide more information on what authors was not able to find or achieve due to the limitation when conducting this study.
Response: As noticed by reviewer, there were clearly defined questions which this manuscript was built upon (1. what constitutes adventure sports and 2. how can they be better understood through an ecological dynamics approach). To clarify the scope of the article, we have not added a distinct limitations section, but we have now added a minor revision to introduction, to emphasize that from the explored existing adventure sport literature, mainly articles from the ecological perspective have been included to remain within the scope of the manuscript, see lines 65-67.
Reviewer 3 Report
Thank you for the opportunity to review this paper. This study addresses a unique topic by proposing a conceptual framework of adventure sport context. This study starts with clarifying the definition and concepts of adventure sport and further discusses constraints, affordances, and form of life in the adventure sport context. This study is well-written and provides a clear concept of adventure sport. However, I would like to ask the authors to consider some suggestions below.
In section 3, it would be better to use a table to summarize the points you address (especially the constraint section).
Moreover, although this study intends to provide theoretical and practical implications, what and how this study can theoretically and practically contribute to extant literature and practice are not clear.
The conceptualization of this study is clear and comprehensive. However, authors need to have sections addressing theoretical and practical implications.
Author Response
Thank you for the opportunity to review this paper. This study addresses a unique topic by proposing a conceptual framework of adventure sport context. This study starts with clarifying the definition and concepts of adventure sport and further discusses constraints, affordances, and form of life in the adventure sport context. This study is well-written and provides a clear concept of adventure sport. However, I would like to ask the authors to consider some suggestions below.
In section 3, it would be better to use a table to summarize the points you address (especially the constraint section).
Moreover, although this study intends to provide theoretical and practical implications, what and how this study can theoretically and practically contribute to extant literature and practice are not clear.
The conceptualization of this study is clear and comprehensive. However, authors need to have sections addressing theoretical and practical implications.
Response: We thank reviewer for constructive and encouraging comments. In line with suggestions provided by reviewer, sections and subheadings have been added to concluding remarks section, to better clarify theoretical and practical implications. The comment about adding a table has been acknowledged. However, since this has not been raised as an issue by other reviewers, we hope the section is in understandable and sufficient form as it reads now, since we have decided not to use a table as suggested.
Round 2
Reviewer 2 Report
Please check the attached file.

Author Response
Subsection "limitations" has now now been included, as suggested by reviewer. Please see lines 765-779.